# Synthesis of Fluorinated and Fluoroalkylated Heterocycles Containing at Least One Sulfur Atom via Cycloaddition Reactions [note 1]

**DOI:** 10.3390/ma15207244

**Published:** 2022-10-17

**Authors:** Grzegorz Mlostoń, Yuriy Shermolovich, Heinz Heimgartner

**Affiliations:** 1Department of Organic and Applied Chemistry, Faculty of Chemistry, University of Lodz, 12 Tamka Street, 91-403 Lodz, Poland; 2Institute of Organic Chemistry, National Academy of Sciences of Ukraine, 5 Murmanska Street, 02094 Kyiv, Ukraine; 3Department of Chemistry, University of Zurich, Winterthurerstrasse 190, CH-8057 Zurich, Switzerland

**Keywords:** fluorinated and fluoroalkylated organic compounds, sulfur heterocycles, organic synthesis, cycloaddition reactions, organic reaction mechanisms

## Abstract

Fluorinated heterocycles constitute an important group of organic compounds with a rapidly growing number of applications in such areas as medicinal chemistry, agrochemicals production, polymer chemistry, as well as chemistry of advanced materials. In the latter case, fluorinated thiophenes are considered as a lead class of compounds with numerous spectacular applications. On the other hand, cycloaddition reactions offer a superior methodology for stereo-chemically controlled synthesis of heterocycles with a diverse ring size and a variable number of heteroatoms. A comprehensive review of methods based on cycloaddition reactions and applied for construction of fluorinated and/or fluoroalkylated S-heterocycles has not yet been published. For this reason, the main goal of the presented review was to fill the existing gap and to summarize the results published over last six decades. In this context, the [3+2]- and [4+2]-cycloadditions (Huisgen reactions, and Diels–Alder reactions, respectively) are of special importance. Some questions related to the discussed mechanisms of cycloaddition processes observed in reactions with electron deficient, fluorinated substrates (dipolarophiles and dienophiles), and electron-rich sulfur containing counter partners, are of fundamental importance for the development of interpretations of organic reaction mechanisms.

## 1. Introduction

In recent decades, we have witnessed the growing importance of fluorinated organic compounds in practically all areas of organic synthesis, and the elaboration of new methods for the preparation of fluorinated heterocycles belongs to the challenging problems of current organic chemistry [1,2]. It is well known that one of the best methods for the construction of the non-aromatic, heterocyclic compounds are cycloaddition reactions, and those most frequently applied are [2+1]-, [2+2]-, [3+2]- and [4+2]-cycloadditions. Starting with properly designed components containing fluorine (thiocarbonyl compounds, carbenes, 1,3-dipoles, dienes), the synthesis of a fluorinated S-heterocycle can be achieved via a cycloaddition step or by further transformation of the initially obtained cycloadduct. The presence of electron-withdrawing fluoroalkyl groups enhance the reactivity of both dipolarophiles and dienophiles. Similarly, fluorinated 1,3-dipoles display high reactivity towards electron-rich dipolarophiles and [3+2]-cycloadditions performed with thiocarbonyl dipolarophiles are of special importance for the construction of fluoroalkylated sulfur heterocycles. For materials chemistry, the development of methods for the synthesis and studies on the reactivity of fluorinated heterocycles are of great interest. In a recent comprehensive review, the methods applied for syntheses of diverse heterocycles bearing fluorine atoms and/or fluoroalkylated substituents were summarized. However, sulfur-containing heterocycles are inadequately represented, mainly by fluorothiazole and fluorobenzothiazole derivatives [3]. The goal of the present work is an overview of the methods which are of practical importance for the preparation of fluorine-containing sulfur heterocycles upon exploration of cycloaddition reactions as basic tools for the construction of the heterocyclic core.

## 2. Three-Membered S-Heterocycles (Thiiranes)

The [2+1]-cycloaddition of difluorocarbene, generated by thermal decomposition of PhHgCF_3_ (Seyfert’s reagent) in boiling benzene, with 2,2,4,4-tetramethyl-3-thioxocyclobutanone (**1a**) and its dithioxo analogue **1b** gave the corresponding spirocyclic *gem*-difluorothiiranes **2a**,**b** (Figure 1) [4]. In the case of **1b**, the formation of two isomeric 2:1-adducts, i.e., *cis*- and *trans*-**3**, in a ratio of 3:1 was also observed.

Aromatic thioketones **1c**,**d** react under the same conditions with: CF_2_ yielding *gem*-difluoroethenes **4** via spontaneous desulfurization of the intermediate thiiranes **2c**,**d** (Figure 2) [4]. The mechanism of the desulfurization process is not known, but it is very likely that CF_2_ can play an important role in this reaction.

The analogous desulfurization of an intermediate *gem*-difluorothiirane, generated from trimethylsilyl 2,2-difluoro-2-fluorosulfonylacetate and dithioesters, led to *gem*-difluorovinylsulfides [5].

Heating of perfluoropropenoxide in the presence of difluorothiophosgene (thiocarbonyl fluoride) or trifluorothioacetyl fluoride in a closed reactor under pressure at 175 °C yielded the corresponding perfluorinated thiiranes **2e** and **2f**, respectively, in fair yields (Figure 1) [6]. The same method was applied for the preparation of chlorotrifluorothiirane (**2g**) [6,7].

In the case of trifluoromethyl dithiochloroformate, the thermal reaction with perfluoropropenoxide leads to the corresponding vinyl sulfide **4c**, formed via spontaneous desulfurization of the intermediate thiirane [8]. In all of these reactions, thermal decomposition of perfluoropropenoxide leads to difluorocarbene as the reactive intermediate.

A superior method for the preparation of fluorinated thiiranes is the 1,3-dipolar electrocyclization (1,3-DE) of transient, fluorinated thiocarbonyl *S*-methanides **7**, generated from the corresponding 1,3,4-thiadiazolines **6** (Figure 3, Table 1). The latter heterocycles are smoothly formed via [3+2]-cycloaddition of a thiocarbonyl compound as dipolarophile with diazomethane derivatives **5**. The fluorine atom or a fluoroalkyl group may originate either from one or from both reaction partners.

Hexafluorothioacetone (**1e**) is known as a highly reactive dipolarophile, which easily reacts with diphenyldiazomethane (**5a**) at a low temperature in pentane solution to yield 2,2-diphenyl-3,3-bis(trifluoromethyl)thiirane (**2h**) after the spontaneous elimination of N_2_ [9]. A similar reaction course with immediate evolution of N_2_ was observed in reactions of thiobenzophenone (**1c**) with methyl α-diazo-3,3,3-trifluoropropanoate (**5b**) [10] as well as thiophosgene or monofluorothiophosgene with 2-diazohexafluoropropane (**5c**) [11].

The sterically crowded cyclobutanethiones **1a**,**h** react with 2,2,2-trifluorodiazoethane (**5d**) in CH_2_Cl_2_ solution at room temperature, and after 24 h, the spirocyclic thiiranes **2l** and **2m**, respectively, were isolated in good yields [12]. However, in analogous reactions stopped just after de-coloration of the mixture (ca. 60 min), the corresponding [3+2]-cycloadducts **6a**,**b** were obtained as relatively stable solids in good yields. In contrast to the cycloaliphatic thioketones **1a**,**h**, thiobenzophenone (**1c**) reacted with **5d** under the same conditions to give 3,3,3-trifluoro-1,1-diphenylpropene as the product of spontaneous desulfurization of the in situ-formed thiirane.

Only in the case of 2,2,5,5-tetrakis(trifluoromethyl)-1,3,4-thiadiazoline (**6a**), formed from 2-diazohexafluoropropane (**5c**) and hexafluorothioacetone (**1e**), was no evolution of N_2_ observed, and this product could be distilled in vacuum without decomposition [13]. The presence of four CF_3_ groups in **6a** results in unusual stability, as in other cases 1,3,4-thiadiazolines are known to eliminate N_2_ easily [14].

## 3. Four-Membered S-Heterocycles

### 3.1. Thietanes

Hexafluorothioacetone (**1e**), existing in an equilibrium with its dimer **8a**, displays high reactivity in [2+2]-cycloadditions with both electron-rich and electron-deficient C=C bonds. Selected examples of these reactions are depicted in Figure 4 [15,16,17,18,19]. The first reactions of that type were reported with vinyl ethers and thioethers, which yielded 4-substituted 2,2-bis(trifluoromethyl)thietanes **9a**,**b** in a regioselective manner [15]. Analogous [2+2]-cycloadditions were performed with *N*-vinylformamide and *N*-vinylcarbazole, leading to products **9c** and **9d**, respectively [18].

In the case of *gem*-disubstituted ethylenes, the corresponding thietanes were obtained with both 1,1-bis(dimethoxy)- and 1,1-bis(methylsulfanyl)ethene as well as with the electron-deficient 1,1-bis(trifluoromethyl)ethene [17]. Whereas the disulfanyl derivative afforded the product **9f** as a stable compound, the dimethoxy analogue gave the opposite regioisomer as the product of the kinetic control, which after three months at 25 °C converted completely to the thermodynamic product **9e**. Unexpectedly, the electron-deficient 1,1-bis(trifluoromethyl)ethene yielded the same type of regioisomer as observed in the case of electron-rich ethenes. The structures of the stable products **9g** [17], **9e** and **9f** [20] were established by X-ray crystallography. In addition, styrene and the in situ-generated norbornadiene are captured be **1e** yielding the corresponding thietane derivatives **9h** and **9i**, respectively [19,21].

The [2+2]-cycloadditions of alkenes and 1,3-dienes with the in situ-generated *S*,*S*-dioxide of **1e**, i.e., bis(trifluoromethyl)sulfene (**10**), afforded thietane-*S*,*S*-dioxides of type **11** (Figure 5). In all cases, these products were formed regio-selectively [22].

In the reactions with 1,3-dienes, such as 2,5-dimethylhexa-2,4-diene and (*E*,*E*)- and (*E*,*Z*)-hexa-2,4-dienes, [2+2]-cycloadditions also governed the formation of four-membered products **11f**,**g**. In the two latter cases, mixtures of *trans*- and *cis*-isomers were formed. The (*E*,*E*)-diene yielded a 96:4 mixture of *trans*- and *cis*-**11g**, whereas the (*E*,*Z*)-diene gave the *cis*-isomer as the major component (18:82).

### 3.2. 1,3-Dithietanes

The perfluorinated thioketone **1e** (hexafluoroacetone) and α,α,α-trifluorothioacetophenone (**1i**) easily undergo head-to-tail dimerization to give 1,3-dithietanes **8a** and **8b**, respectively (Figure 6) [17,23]. In both cases, the thioketones are prepared from corresponding starting materials and dimerize spontaneously in the reaction mixture.

The analogous tendency for the [2+2]-cycloaddition leading to the 1,3-dithiethane derivative **8c**, formed as an 85:15 mixture of *trans*- and *cis*-isomer, was reported for *N*-acetyl trifluoroacetic thioamide (**12**) [24]. Some examples of the use of dithietane **8a** in organic synthesis were recently described in a review [25].

The reaction of thioketones or alkenes with bis(trifluoromethyl)thioketene (**13**) results in the regioselective formation of 1,3-dithietanes **14** (Figure 7) or thietanes, correspondingly [26].

## 4. Five-Membered S-Heterocycles

### 4.1. Thiolanes (Tetrahydrothiophenes)

The thiolane ring constitutes a core fragment in many compounds demonstrating diverse types of biological activities and they have found numerous therapeutic applications [27,28,29]. In spite of numerous commercial offers for their supply in large quantities, there are only a limited number of reports available on efficient laboratory synthesis of the 2- or 3-trifluoromethylated thiolanes. Obviously, in this situation the studies that are focused on the exploration of the cycloaddition methodology for the synthesis of this class of S-heterocycles remain the most relevant.

An attractive approach to the synthesis of fluoroalkylated thiolanes is the use of electron-rich, *S*-centered 1,3-dipols known as thiocarbonyl ylides (thiocarbonyl *S*-methanides) **15**. They have to be generated in situ, preferably by thermal decomposition of corresponding 1,3,4-thiadiazolines. In the presence of a suitable dipolarophile, e.g., fluorinated α,β-unsaturated ketones **16**, they easily undergo [3+2]-cycloaddition leading to five-membered *S*-heterocycles **17** (Figure 8). If such dienophiles belong to the group of electron-deficient ethylenes, they are prone substrates for the [3+2]-cycloadditions with **15** [30].

The new 4-trifluoromethyl thiolanes **18** containing an ester, sulfone, sulfonamide, sulfoximine or phosphonate moiety at C(3) were prepared by [3+2]-cycloaddition reactions of 3,3,3-trifluoropropene derivatives with the parent thiocarbonyl ylide **19** generated in situ from chloromethyl trimethylsilyl sulfide (Figure 9) [31].

Recently, a new method for the synthesis of 2-trifluoromethyl thiolanes **21**, based on the transformation of the unsaturated six-membered sulfoxide **20**, was elaborated. The latter compound was prepared by the [4+2]-cycloaddition of the corresponding dithiocarboxylate with a suitable 1,3-diene. Taking into account the high total yield of isolated products, this ring contraction may be considered as a method of choice for the preparation of 2-trifluoromethyl thiolanes **21** (Figure 10) [32].

### 4.2. 1,3-Dithiolanes

Middleton and Sharkey observed the formation of 2,2,4,4-tetrakis(trifluoromethyl)-1,3-dithiolane (**22**) when hexafluorothioacetone was treated with diazomethane in pentane solution at −78 °C and subsequently the reaction mixture was rapidly distilled (Figure 11) [9].

Apparently, 1,3,4-thiadiazoline **23**, formed at a low temperature, is not a stable cycloadduct and spontaneously eliminated nitrogen to generate hexafluorothioacetone *S*-methanide (**24**), which was in situ-trapped by the starting thioketone yielding **22** in a regioselective manner; formation of the isomeric 2-CH_2_ derivative was not observed. Another, colorless minor product with the molecular formula C_14_H_4_F_24_S_4_ was also reported, but its structure could not be elucidated.

Some *N*,*N*-disubstituted thioamides containing polyfluoroalkyl substituents enter [3+2]-cycloaddition with the parent thiocarbonyl *S*-methanide (**19**) and new 4-polyfluoroalkyl-1,3-dithiolanes **25** were obtained in these reactions in good to high yields (Figure 12, Equation (**A**)) [33].

The regioisomeric 1,3-dithiolanes of type **26** and **27** were synthesized via a domino reaction initiated by thermal N_2_ extrusion from the CF_3_-substituted 1,3,4-thiadiazole **28**, and the in situ-generated, reactive thiocarbonyl ylide **29** (X = CO) trapped thioketones **1k**, **l** yielding mixtures of **26** and **27** in favor of the sterically less hindered 5-CF_3_ isomer **26** (Figure 12, Equation (**B**)) [12,34].

The 2,2,2-trifluorodiazoethane easily reacts with cycloaliphatic and aromatic thioketones forming corresponding 1,3,4-thiadiazolines of type **28**, which were subsequently used as precursors of fluorinated thiocarbonyl *S*-ethanides **29** [12]. For example, 2,2,4,4-tetramethyl-3-thioxocyclobutanone (**1a**) reacts with trifluorodiazoethane to form 1,3,4-thiadiazoline **28a**, which, after elimination of N_2_ at room temperature, serves as the source of **29** (X = CO). This intermediate is efficiently trapped with thiobenzophenone to give, in contrast to the corresponding *S*-methanide, the less hindered 1,3-dithiolane **30a** as the sole cycloadduct (Figure 12).

An unexpected formation of 1,3-dithiolanes **31** was observed upon treatment of perfluoropropene with elemental sulfur in the presence of vinyl *O*-alkyl ethers in DMF solution at 45–65 °C using CsF as a catalyst. Under the same conditions, dihydrofuran gave the bicyclic 1,3-dithiolane **32** in a 58% yield as a mixture of *cis*- and *trans*-isomer (Figure 13) [17].

### 4.3. 1,3-Dithioles

Replacement of the electron-rich vinyl ethers in the CsF-catalyzed sulfurization reactions of perfluoropropene by the electron-deficient dimethyl acetylenedicarboxylate (DMAD) led to the corresponding 1,3-dithiole **33** in a 70% yield (Figure 14) [17].

The formation of both 1,3-dithiolanes **31**/**32** and 1,3-dithiole **33** in the fluoride anion-catalyzed sulfurization of perfluoropropene was explained via [3+2]-cycloaddition of in situ-generated 3,3-bis(trifluoromethyl)dithiirane **34** onto the C=C and C≡C bond, respectively (Figure 15). It is very likely that these cycloaddition reactions comprise the interception of an intermediate diradicaloid **35**, generated by homolytic cleavage of the S–S bond of the congested dithiirane ring (Figure 15) [17].

### 4.4. 1,2-Dithiolanes

The 3,3-bis(trifluoromethyl)-5-alkoxy-1,2-dithiolanes **36** were prepared by the CsF-initiated reaction of 4-alkoxy-2,2-bis(trifluoromethyl)thiethanes **9a** (see Section 3.1) with elemental sulfur in DMF solution (Figure 16) [16,17].

The fluoride anion is supposed to open the four-membered ring of the starting **9a** yielding the thiolate anion as an intermediate, which subsequently reacts with another molecule of **9a**. Ring closure of the postulated intermediate occurs with elimination of an olefinic side product and formation of the 1,2-dithiole derivative **36** in yields between 73 and 87% [17].

In a recent publication, an alternative mechanism for the formation of 1,2-dithiole derivatives via sulfurization of 2,3-diarylcyclopropenthiones **37** with elemental sulfur in the presence of catalytic amounts of tetrabutylammonium fluoride (TBAF) was formulated to explain the role of the fluoride anion as an activator. This mechanism corresponds to a formal [2+3]-cycloaddition of elemental sulfur (as S_2_) with the congested three-membered ring (Figure 17) [35].

Similar reactions of 2,3-diarylcyclopropenethiones, without explanation of the crucial role of the fluoride anion, have also been published [36,37]. It has to be stressed that the fluoride anion-mediated sulfurization of 2,3-diarylcyclopropenthiones **37** should be considered as a useful method for the synthesis of 1,2-dithiole-3-thiones of type **38**. The latter sulfur heterocycles are of importance as biologically active compounds, and in a recent review, the method of their synthesis as well as diverse transformations have been summarized [38].

### 4.5. 1,3-Oxathiolanes

Thiocarbonyl *S*-methanides **15**, generated in situ by N_2_ elimination from 1,3,4-thiadiazolines (see Section 4.1), undergo regioselective [3+2]-cycloaddition with the activated carbonyl group of trifluoropyruvic methyl ester to give 1,3-oxathiolanes **39a** in a good yield (Figure 18) [10].

In contrast to the α,β-unsaturated ketones **16** (Figure 8), the isomeric trifluoromethyl ketones **16′** react with thiocarbonyl ylides **15** in a chemo- and regioselective manner via [3+2]-cycloaddition with the carbonyl group to yield 5-styryl-5-trifluoromethyl-1,3-oxa-thiolanes **39′** (Figure 18) [30].

### 4.6. 1,3,4-Thiadiazole Derivatives

A new and accessible method for the synthesis of 2-amino-3-phenyl-2-polyfluoroalkyl-2,3-dihydro-1,3,4-thiadiazolines **40** containing various functional groups at C(5) via [3+2]-cycloaddition reactions of polyfluoroalkane thioamides **41** with fluorinated nitrile imines **42a** was recently described (Figure 19, equation above) [39,40]. Regioselective [3+2]-cycloaddition of 1,3-dipoles **42b** with aryl, hetaryl and ferrocenyl thioketones [41,42] as well as with 2,3-diphenylcyclopropenethione [43] lead to the formation of a new type of fluoroalkylated 2,3-dihydro-1,3,4-thiadiazoles **43** and **44**, respectively.

Analogously, fluorinated nitrile imines **42b** react with monomeric aryl/hetaryl substituted thiochalcones yielding 2-styryl-substituted 1,3,4-thiazolidines **45** in a chemo- and regioselective manner (Figure 19, equation below) [44].

Remarkably, cycloadditions of non-fluorinated nitrile imines with thiochalcones have not been studied to the date. However, chalcones were reported to react with non-fluorinated nitrile imines upon selective involvement of the C=C bond in [3+2]-cycloaddition reactions leading to pyrazole derivatives [45,46].

An interesting and rather unexpected observation was made in the course of studying the [3+2]-cycloaddition of thiocarbonyl ylide **46** to 1,2-bis(trifluoromethyl)-ethene 1,2-dicarbonitrile. Along with the expected five-membered product, i.e., the thiolane **47**, formation of a seven-membered thiazepine derivative **48**, with a ketenimine fragment incorporated into the heterocyclic ring, occurred predominantly (ratio 87:22) (Figure 20) [47]. Its reaction with methanol gave the spirocyclic thiazepine derivative **49** as a mixture of two diastereoisomers.

Other sterically crowded thiocarbonyl *S*-methanides that were derived, e.g., from 2,2,6,6-tetramethylcyclohexanethione, reacted with electron deficient, fluorinated ethylenes analogously, yielding seven-membered products. Some of them were isolated as crystalline materials and their structures, as well as that of some products of their further conversions, were unambiguously confirmed by X-ray measurements [48,49]. The formation of these unusual *N*,*S*-heterocycles was explained by the non-concertedness of the expected cycloaddition process and formation of stabilized zwitterionic intermediates of type **50**, which competitively can undergo either 1,5- or 1,7-cyclization yielding the five-membered thiolane **47** or the seven-membered 1,3-thiazepine **48**, respectively. The unusual stability of the heterocumulene (ketenimine) unit –N=C=C– was rationalized by ‘the magic effect’ of the trifluoromethyl group located at the neighboring C-atom [50]. Some chemical properties of **48** will be described in Section 6.

### 4.7. 1,4,2-Oxathiazoles

The 3-trifluoro- or difluoromethylated 1,4,2-oxathiazoles **51** can be efficiently prepared via the regioselective [3+2]-cycloaddition of fluorinated nitrile oxides **52** with thioketones **1** (Figure 21) [51]. The nitrile oxides **52** are generated in situ by treatment of hydroximoyl bromides **53**, which are obtained conveniently in two steps from the corresponding aldehyde derivatives **54**, with triethylamine [52].

## 5. Six-Membered S-Heterocycles

### 5.1. Thiopyran Derivatives

Synthetic methodologies based on the hetero-Diels–Alder reaction are widely employed in organic chemistry. Using heterodienes or heterodienophiles in the [4+2]-cycloaddition reaction makes it possible to construct complex natural products or their analogues containing a six-membered heterocyclic framework. Thiocarbonyl compounds are well-known representatives of heterodienophiles which, for example, found applications for the preparation of thioglycoside derivatives [53] or thiashikimic acid [54]. Electron-withdrawing groups in α-position to the thiocarbonyl group lower the LUMO energy of the heterodienophile and facilitate the cycloaddition. Therefore, the polyfluoroalkyl thiocarbonyl compounds are excellent dienophiles and can be used for the preparation of diverse sulfur-containing heterocycles.

So far, the [4+2]-cycloaddition reactions of polyfluoroalkyl thiocarbonyl compounds with 1,3-dienes remain the main method for the synthesis of fluorine-containing derivatives of thiopyrans. Fluorinated thioaldehydes, thioketones, esters, amides, and halides of polyfluoroalkyl thionocarboxylic acids are used as thiocarbonyl compounds.

The extremely reactive polyfluoroalkane-derived thioaldehydes were synthesized, and their existence was proved by spectroscopic methods and their Diels–Alder reactions [55,56,57]. The trifluorothioacetaldehyde (**55a**) as well as the thioketones CF_3_C(S)R (R = Me, Ph) could be obtained in good yields by flash vacuum pyrolysis of 1,3-dithiolane-1,1-dioxides (Figure 22, equation shown above) [23,55]. The thioaldehydes **55b**,**c** with longer perfluoroalkane chains were prepared by the reactions of the corresponding aldehydes with trimethylthiophosphates [56]. Various Diels–Alder products **56**–**60** were prepared by the reactions of these thioaldehydes as well as of polyfluoroalkyl-substituted sulfines **61** (Figure 22, equation shown below), and the stereochemical considerations were presented in the published articles [23,57,58].

Aromatic thioketones, such as thiofluorenone (**1k**) and thiobenzophenone (**1c**) and its derivatives, were demonstrated to react with cyclopentadiene and other cyclic and acyclic 1,3-dienes at room temperature yielding the expected bicyclic thiopyrans **62** (Figure 23). The kinetic measurements demonstrated ‘superdienophilic’ reactivity for both thioketones and structurally similar selones [59]. Remarkably, polyhalogenated cyclopentadienes bearing chlorine and fluorine substituents definitely reacted slower and the studied [4+2]-cycloadditions could be performed at room temperature only with the ‘superdipolarophilic’ thiofluorenone (**1k**). Its reaction with 1,2,3,4-tetrachloro-5,5-difluorocyclopentadiene leading to thiopyran derivative **63** is depicted in (Figure 23) [60].

A simple procedure for the preparation of new Diels–Alder adducts of different polyfluorinated thioketones **64** was elaborated recently (Figure 24) [61,62]. The corresponding adducts of type **65** (e.g., **65a**–**f**) have been prepared in a 30–78% yield by the reaction of perfluorinated olefins, sulfur and 1,3-dienes in the presence of CsF acting as a catalyst.

The regiochemistry of the Diels–Alder reaction of open-chain 1,3-dienes with hexa-fluorothioacetone, generated in situ by thermal, CsF-catalyzed decomposition of its dimer 2,2,4,4-tetrakis(trifluoromethyl)-1,3-dithietane (**8a**), was studied recently and the results were discussed in detail [63].

The pioneering works of W. Middleton described the first examples of the Diels–Alder reactions of trifluoroacetyl fluoride [64] and perfluorinated thioketones [65] with 1,3-dienes. These reactions were studied in more detail much later. The cycloaddition of the chloride and fluoride of polyfluoroalkane thiocarboxylic acids with 1,3-dienes proceeded rapidly at 0 °C. The stability of the cycloadducts formed depended on the length of the polyfluoroalkyl chain. Trifluorothioacetyl chloride afforded a relatively stable adduct **66**, which was isolated and characterized spectroscopically (Figure 25). Chlorides with longer chains gave directly 2*H*-thiopyrans **67** after the evaporation of the reaction mixture. Dehydrochlorination of the CF_3_-substituted thiopyran was achieved only after heating at 100 °C [66]. The thiopyrans obtained turned out to be convenient starting compounds for the 2*H*-thiopyrans **67** preparation of the first 2-polyfluoroalkyl-substituted thiopyrylium salts, which in turn can efficiently be used for the synthesis of fluorinated thiopyranosides and nucleosides [66].

Another interesting synthetic application of fluorinated thiopyran derivatives **66** consists in the formation of 2-polyfluoroalkylthiophenes **69**/**70** via the sulfur-assisted ring contraction of 4,5-dibromo-2-chloro-2-(tri- or difluoromethyl)tetrahydrothiopyrans **71** (Figure 26) [67].

Over the past two decades, the factors affecting the reactions of polyfluoroalkane thiocarboxylic acid esters with 1,3-dienes of various structures have been studied in detail and diverse synthetic applications of the obtained derivatives of dihydrothiopyrans have been considered [32,68,69,70]. As a result of these reactions, at least one new stereogenic center was generated by the Diels–Alder addition. Therefore, the application of a proper chiral thionoester could influence the stereochemical outcome of the cycloaddition and could be used in the construction of optically active compounds.

The preparation of a series of thionoesters **72** with various optically active substituents, which can serve as chiral auxiliaries in asymmetric syntheses, allowed the observation of the first examples of asymmetric induction in the thia-Diels–Alder cycloaddition involving polyfluoroalkane thionocarboxylates, which provided 2-fluoroalkyl-2-alkylsulfanyl-3,6-dihydro-2*H*-thiopyrans **73** in good yields but a low to modest *de* of 6–60% (Figure 27, Table 2) [70].

The influence of the nature of the diene and dienophile and the reaction conditions on the asymmetric induction of the cycloaddition have been examined. It has been found that electronic factors have a minimal effect on the stereoselectivity of the cycloaddition. Quantum chemistry (DFT) calculations indicate that the differences in the activation energies are larger than the relative energies of the cyclic adducts. This result allows the conclusion that the stereoselectivity of the formation of thiopyrans **73** is kinetically driven: the observed *de* refers to different free-activation energies inherent to the corresponding transition states.

Much less is known about the reactions of derivatives of polyfluoroalkane thionocarboxylic acid such as the *O*-esters and amides. However, even the available data allow a conclusion to be drawn about the significant effect of the nature of the heteroatom in the Alk_F_C(S)XR (X = S,O,N) on the rate of the cycloaddition reaction. Thus, in the case of dithioethers **72a**–**d**, the reaction with 2,3-dimethylbuta-1,3-diene proceeds at room temperature, while a similar reaction with the ester **72e** requires vigorous heating [70].

Alkylamides of polyfluoroalkane thiocarboxylic acids **74a**–**d** do not react with 1,3-dienes even after many hours of heating in a sealed ampoule. A successful reaction to give 3,6-dihydro-2*H*-thiopyrans **75a**–**d** was ensured only by microwave activation in *N*-methylpyrrolidone (NMP) solution [71] or using an N-acylated thioamide derivative **76** (Figure 28) [69]. The positive results observed in the last cases, leading to products **77**, was explained by the electron-withdrawing influence of the amide substituent on the thio-carbonyl group.

The high activity of hexafluorothioacetone in cycloaddition reactions with 1,3-dienes is explained by the influence of the electron-withdrawing trifluoromethyl groups. In the case of thioacid derivatives, the opposite effect is observed. The electronegativity of the nitrogen and oxygen atoms is significantly higher than that of sulfur, but the rates of cycloaddition reactions are lower. Perhaps the reason for this unexpected effect may be the interaction of the sulfur-carbon multiple bond orbitals with the orbitals of oxygen or nitrogen heteroatoms, which is not so significant in the case of the sulfur atom due to the larger size of the latter.

### 5.2. 1,4-Dithiines, 1,4-Dithianes and 1,2,4,5-Tetrathianes

Syntheses of fluorine-containing derivatives of 1,4-dithiane by cycloaddition reactions are only known in two examples. In an early work, it was suggested that the formation of 1,4-dithiine **78** and 2,3-dihydro-1,4-thiines **79** as the result of the reaction of bis(trifluoromethyl)dithietane **80** to acetylenes and olefins occurs according to the Diels–Alder reaction of the intermediate bisthiocarbonyl compound **81**, which is formed via a preliminary cleavage of the sulfur–sulfur bond of **80** (Figure 29) [72].

The 2,3-bis(benzylthio)-2,3-bis(octafluoropentyl)-1,4-dithiane **82** was obtained by dimerization of thiocarbonyl *S*-methanide **83** generated in the reaction of dithioester **84** with diazomethane (Figure 30) [73]. The characteristic structural feature of this compound is the *cis*-configuration of the fluoroalkyl substituents in the six-membered ring.

The thermal reaction of bis(trifluoromethyl)sulfine (**85**) with thiophosgene at 110 °C leads to 3,3,6,6-tetrakis(trifluoromethyl)-1,2,4,5-tetrathiane (**86**) as a minor product (3%) resulting from the thermal decomposition of the initially formed sulfenylchloride **87**, which in this case was isolated as the major product (Figure 31) [74].

The decomposition of 2-diazohexafluoropropane (**88**) in carbon disulfide at 150–175 °C produced the tetrathiane **89** in trace amounts, presumably formed via the addition of bis(trifluoromethyl)carbene to carbon disulfide and the subsequent dimerization of the adduct (Figure 31) [26].

### 5.3. 6H-1,3,4-Thiadiazines

Substituted thiobenzophenones **1c**,**l**–**o** were reacted with 3,6-bis(trifluoromethyl)-1,2,4,5-tetrazine (**90**), and in contrast to other dienes studied by Sauer et al., these reactions had to be performed in boiling toluene at 100 °C. Under these conditions, thiofluorenone (**1k**) underwent decomposition and for this reason it could not be used in the studied hetero-Diels–Alder reactions with **90**. The initially formed [4+2]-cycloadducts **91** with thiobenzophenones could not be isolated and the target products, 6*H*-1,3,4-thiadiazines **92**, were formed in situ after the spontaneous extrusion of nitrogen (Figure 32) [60].

A similar course of hetero-Diels–Alder reaction was observed when 1,2,4,5-tetrazine **90** was treated with alkyl thioformates in boiling toluene. The desired 6*H*-1,3,4-thiadiazines **93** were obtained as stable compounds and could be distillated yielding analytically pure materials (Figure 33) [75]. In contrast, under the same conditions, the reaction of **90** with thioformamides led to 4-aminopyrazoles **94a** as the sole products in ca. 20% yield. The initially formed [4+2]-cycloadducts, 6*H*-1,3,4-thiadiazines **93b**, underwent an unexpected ring contraction to give the bicyclic thiirane **95a**, followed by elimination of sulfur. The final products were identified as pharmacologically interesting 3,5-trifluoromethyl-4-aminopyrazoles **94a** (Figure 33) [75].

In an analogous manner, 6*H*-1,3,4-thiadiazine **93c**, formed via [4+2]-cycloaddition/N_2_-elimination mechanism from in situ-generated thiobenzaldehyde and bis(trifluoromethyl)tetrazine **90**, was converted into 4-phenyl-3,5-bis(trifluoromethyl)pyrazole **94b** (Figure 34) [75].

## 6. Seven-Membered, Sulfur-Containing Heterocycles

Information on the syntheses of sulfur- and fluorine-containing seven-membered heterocycles via a stepwise [4+3]-cycloaddition reaction is fragmentary. The first example of such a reaction, leading to a trifluoromethylated, seven-membered *N*,*S*-heterocycle, was discussed in Section 4.6 (Figure 20).

The ketenimine fragment in the heterocycle **48** easily reacts with nucleophiles, yielding new types of 1,3-thiazepine derivatives **49**, **96** and **97** [76]. In addition, it enters [2+2]-cycloaddition reactions with vinyl ethers and the [3+2]-cycloaddition with diazomethane to give fused heterocycles **98**/**99** and **100**/**101**, respectively (Figure 35) [77]. In the latter cases, the reaction proceeds non-chemoselectively via the addition of CH_2_N_2_ onto the C=C and C=N double bond to give the primary, non-isolable cycloadducts **102** and **103**, which in situ eliminate N_2_ and HF, respectively, to give the final products.

A surprising dimerization of the ketenimine **48b** occurred in acetonitrile at room temperature in the presence of KF as a catalyst. The formation of two diastereoisomers of product **104** in a ratio of 1:1 was explained by the fluoride ion-initiated reaction mechanism via intermediate **105** (Figure 36) [78].

The assumption of the likely [4+3]-cycloaddition reaction of intermediately formed thioketone *S*-sulfide (thiosulfine) **106** with elemental sulfur S_8_ has been suggested by the authors of [79] while studying the reaction of hexafluoroacetone hydrazone (**107**) with disulfur dichloride. The expected bis(trifluoromethyl)hexathiacycloheptane (**108**) was only isolated in a 3% yield (Figure 37).

## 7. Conclusions and Outlook

The biological utility of fluorinated organic compounds [80], including sulfur hetero-cycles, e.g., thiolanes [32,81] and benzothiazoles [82], is demonstrated by recently published reviews and original works. Without a doubt, fluorinated and fluoroalkylated thio-phenes are the most prominent S-heterocycles, widely applied in diverse areas of materials chemistry [83,84,85,86]. On the other hand, some fluorinated S-heterocycles, e.g., *S*-(trifluoromethyl)dibenzothiophenium salts (so called “Umemoto reagents”), are of great importance for the current organic synthesis as fluorinating/fluoroalkylating reagents [87]. In general, fluorine-containing S-heterocycles have rarely been prepared by the direct fluorination/fluoroalkylation of the parent systems and one of the important issues is the regioselectivity of these processes, which complicates the preparation of pure products. For all these reasons, the exploration of highly selective cycloaddition reactions, based on thiocarbonyl dipolarophiles or dienophiles, known as superdipolarophiles [88] and superdienophiles [60], respectively, offer a perfect methodology for the solution of this problem. Thiocarbonyl compounds are also perfect trapping reagents for carbenes in [2+1]-cycloadditions leading to fluorinated thiiranes or their desulfurized derivatives [4]. Remarkably, difluorocarbene reacts with the C=S bond via carbophilic attack, and in these reactions, it resembles the nucleophilic dimethoxycarbene [89] and not the electrophilic dichloro- or dibromocarbene [90].

The present review is aimed at presenting for the first time the synthetic potential of diverse cycloaddition reactions in practical applications for the selective preparation of fluorinated and fluoroalkylated S-heterocycles from three-membered thiiranes to seven-membered *N*,*S*-rings. In addition, the review should be considered as a supplement to the two, recently published reviews on the [4+2]-cycloadditions performed with fluorinated dienes or dienophiles leading, via multi-step mechanisms, to aromatic/heteroaromatic six-membered carbo- and heterocyclic products [91], as well as on the thia-Diels–Alder reactions based on the exploration of sulfur-containing reagents [92].

## Data Availability

Not applicable.

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
