# Peer review of "Synthesis of Fluorinated and Fluoroalkylated Heterocycles Containing at Least One Sulfur Atom via Cycloaddition Reactions†"

_materials, 2022, doi:10.3390/ma15207244_

Round 1

Reviewer 1 Report

The authors presented a very informative review on the synthesis of fluorine-containing heterocycles containing sulfur atoms via cycloaddition reactions. Literature on sulfur-containing heterocycles has been collected and thoroughly analyzed, beginning with three-membered thiiranes and ending with various seven-membered heterocycles.

The review deserves publication in the journal Materials after the following shortcomings have been corrected.

1.      It is necessary to describe the formation of difluorocarbene in more detail on page 4 (lines 82-83); also in the heading for scheme 2, its name is present, but its structure is absent in the scheme itself.

2.      The heading for Scheme 12 describes the content of only the two lower reactions.

3.      Chapter 4.4., which describes the synthesis of 1,3-dithiolanes, lacks a number of interesting transformations, which are described, for example, in a recent review in the journal Molecules (O. A. Rakitin, Molecules, 2021, 26, 3595.).

4.      A number of typos on lines 260, 503, 562, 708, 765 should also be corrected.

Reviewer 2 Report

This manuscript submitted by Mlostoń, Shermolovich and Heimgartner nicely summarizes synthetic accesses to fluorinated sulfur containing heterocycles. The review is justified and the content is presented in a logical order. The authors do a good job of revealing relative reactivities where appropriate and of demonstrating where by-products arise from stray side-reactions. The review authors have contributed significantly to the area, yet the review includes substantial chemistry from other researchers. Hence there is an appropriate selection of content in the manuscript.

This manuscript is certainly publishable and I have a significant general concern and a collection of English improvements/errors. I urge the authors and editor to take them into consideration prior to publication.

General:

1.     Whenever I think of ‘materials’, I guess I think of ‘functional materials’. Indeed, this seems also to define the scope of this journal which reads “To publish research related to all classes of materials including ceramics, glasses, polymers (plastics), composites, semiconductors, magnetic materials, biological and biomimetics materials, silica, dots, and carbon materials, metals, and alloys from nanoscale to bulk. All kinds of functional materials used for the development of medical implants in medicine and in dentistry, coatings and films, pigments, ionic crystals, covalent crystals, metals, and intermetallics are also considered.” Hence although this manuscript has qualities suitable for publication, I question whether it contains sufficient ‘materials’ chemistry to be captured under the scope of the journal. I would suggest there are other MDPI titles where this review might be more applicable.

Suggested specific improvements:

11.     In Scheme 13, is compound 32 missing an oxygen?

22.     The ‘equation above’ and equation below’ referrals of Scheme 12 would better and more clearly be presented with ‘A’ and ‘B’ headings.

33.     I feel the paragraphs that begin on line 245 and on line 250 should have their sequence reversed.  The discussion about the formation of 28 should be the first item presented. Both paragraphs should then be reviewed to minimize repetition.

44.     The description of the chemistry pertaining to Scheme 16 says the reaction is catalyzed by fluoride. This is not true, by definition, as the fluoride is incorporated in one the product.   “Initiated” may be better here.      Similarly, for Scheme 36 and its explanation.

55.     Is the right-most resonance structure important to the message of Scheme 17? I would suggest it is a very minor contributor, and its presence is not required to help understand the chemistry.

66.     Line 312 begins with “In contrast…”, but I do not see a contrasting reaction. I believe “similarly” would be better here.

77.     In scheme 22,    HS=P(OMe)3 does not belong, Should it be   - O=P(OMe)3   ?

88.     On line 421, ‘pioneer’ should be ‘pioneering’

99.     On line 452     ‘allowed to observe’  should be ‘allowed the observation of’. The sentence beginning on line 463 has a similar problem.

110. There are a couple minor spelling issues that can be addressed by editorial staff.

Author Response

Plase see the atachement.

Reviewer 3 Report

 Dear authors,

The manuscript entitledSynthesis of fluorinated and fluoroalkylated heterocycles containing at least one sulfur atom via cycloaddition reactions” disclosed some interesting, new results on heterocycles derivatives. I agree with its publication after minor revisions. The subject of this manuscript is interesting and is within the scope of the journal but it is necessary to pay attention to some points.

1-  Please check the spelling of words. For example: In line 13, the first line of the abstract, please correct “organic counds”; in line 465, “the observed de refers”; in line 455 “to modest de of”;

2-  I suggest adding the biological activities of these compounds in the introduction because of their importance.

3-  In some schemes, intermediates are not shown. If there are, it is better to mention them.

Author Response

Please see the atachement.
